# Generating Heterokaryotic Cells via Bacterial Cell-Cell Fusion

Shraddha Shitut,[a,b,c] Meng-Jie Shen,[b] Bart Claushuis,[c] Rico J. E. Derks,[d] Martin Giera,[d] Daniel Rozen,[c] Dennis Claessen,[c] Alexander Kros[b]

[a]Origins Centre, Groningen, the Netherlands
[b]Department of Supramolecular & Biomaterials Chemistry, Leiden Institute of Chemistry, Leiden University, Leiden, the Netherlands
[c]Institute of Biology, Leiden University, Leiden, the Netherlands
[d]Center for Proteomics and Metabolomics, Leiden University Medical Center, Leiden, the Netherlands

**ABSTRACT** Fusion of cells is an important and common biological process that leads to the mixing of cellular contents and the formation of multinuclear cells. Cell fusion occurs when distinct membranes are brought into proximity of one another and merge to become one. Fusion holds promise for biotechnological innovations, for instance, for the discovery of urgently needed new antibiotics. Here, we used antibiotic-producing bacteria that can proliferate without their cell wall as a model to investigate cell-cell fusion. We found that fusion between genetically distinct cells yields heterokaryons that are viable, contain multiple selection markers, and show increased antimicrobial activity. The rate of fusion induced using physical and chemical methods was dependent on membrane fluidity, which is related to lipid composition as a function of cellular age. Finally, by using an innovative system of synthetic membrane-associated lipopeptides, we achieved targeted fusion between distinctly marked cells to further enhance fusion efficiency. These results provide a molecular handle to understand and control cell-cell fusion, which can be used in the future for the discovery of new drugs.

**IMPORTANCE** Cell-cell fusion is instrumental in introducing different sets of genes in the same environment, which subsequently leads to diversity. There is need for new protocols to fuse cells of different types together for biotechnological applications like drug discovery. We present here wall-deficient cells as a platform for the same. We identify the fluidity of the membrane as an important characteristic for the process of fusion. We demonstrate a cell-specific approach for fusion using synthetically designed peptides yielding cells with modified antibiotic production profiles. Overall, wall-deficient cells can be a chassis for innovative metabolite production by providing an alternative method for cell-cell fusion.

**KEYWORDS** cell fusion, wall deficiency, coiled-coil peptides, heterokaryon, cell membranes, cell wall deficient, lipopeptides, membrane fluidity, protoplast fusion

Cell fusion has been studied in many different eukaryotic cell types (1) and is crucial for tissue repair and regeneration, phenotypic diversity, viral transmission, and recombination (2). The process of fusion proceeds via several steps: cell adhesion, recognition of cell surface components, membrane remodeling, and in some cases nuclear fusion (3). These processes are highly influenced by lipid-lipid interactions (4), which have been studied using coarse-grained lipid models and lipid vesicles (5, 6). Fusion in eukaryotic cells is induced via SNARE proteins that form complexes to bridge together membranes by pulling cells close to each other (7). The potential for SNARE proteins, or related tools that bridge membranes, to facilitate bacterial fusion has not yet been explored.

For fusion between bacteria, the protective cell wall surrounding the cells has to be removed by treatment with lysozyme and/or cell wall-targeting agents. Fusion is then

Address correspondence to Shraddha Shitut, shraddha.shitut@gmail.com, Dennis Claessen, d.claessen@biology.leidenuniv.nl, or Alexander Kros, a.kros@chem.leidenuniv.nl.

The authors declare no conflict of interest.

induced via chemicals like poly(ethylene glycol) (PEG) (8–10) or electric charge (11). Recent evidence indicates that many bacteria can transiently shed their cell wall when exposed to environmental stressors (12). When these stressors are removed, wall-deficient cells can rebuild their cell wall and revert to their walled state. Alternatively, prolonged exposure to these stressors can lead to the formation of so-called L-forms, which can efficiently propagate without their wall (13–16). Much like lipid vesicles, L-form growth and division is regulated by physicochemical forces that deform the cell membrane, leading to an irregular assortment of progeny cells. This makes them suitable to understanding the dynamics and consequences of cellular fusion, as well as to identifying factors that affect this process.

In this study, we showed that fusion induced between L-form cells is a dynamic process whose frequency is dependent on the age of the bacterial culture; this, in turn, is determined by the fluidity of the cell membrane, which we confirmed by chemically manipulating membrane fluidity. In addition, we demonstrate for the first time that complementary lipidated coiled-coil peptides (a synthetic mimic inspired by SNARE proteins) increase the efficiency and specificity of cell-cell fusion. Importantly, fusants resulting from this process are viable and express markers from both parental chromosomes. This opens avenues to designing complex heterokaryotic or hybrid cells that have potential not only to answer questions on evolution of complexity but also to enable novel applications in biotechnology.

## RESULTS

**A dual marker system for identifying cell-cell fusion.** To study cell-cell fusion, we created two fluorescent strains by integrating plasmids pGreen or pRed2 into the *attB* site in the genome of an L-form derivative of the actinobacterium *Kitasatospora viridifaciens* (Fig. 1A). The strain carrying pGreen constitutively expresses enhanced green fluorescent protein (EGFP) and is apramycin resistant, while the strain carrying pRed2 constitutively expresses mCherry and is hygromycin resistant (Fig. 1A). We first confirmed resistance to these antibiotics by determining the susceptibility of each strain to both antibiotics (Fig. 1B and E). The strain expressing resistance to apramycin (referred to as AG [for apramycin-green]) was able to grow at 50 $\mu$g mL$^{-1}$ apramycin. The strain that was hygromycin resistant (referred to as HR [for hygromycin-red]) could grow at 100 $\mu$g mL$^{-1}$ hygromycin. Resistance to one antibiotic did not provide cross-resistance to the other. Confirmation of the fluorescence reporters was obtained via microscopy with cytoplasmic EGFP detected in the AG strain and mCherry detected in the HR strain, with no bleed-through in the other channel (Fig. 1C).

**Fusion of L-forms using centrifugation and PEG.** L-forms show a structural resemblance to protoplasts, which are often used for genome reshuffling in plants and bacteria via the process of cell-cell fusion. After fusion, these protoplasts can revert back to their walled state. To analyze the ability of L-forms to fuse, we tested some commonly used methods for protoplast fusion (9, 17, 18), namely, mechanical force-induced fusion (centrifugation) and chemically induced fusion (with PEG) (Fig. 1D). Nonspecific fusion between AG and HR strains via centrifugation or PEG could result in three different genotypes: AG/HR, AG/AG, and HR/HR. However, genetically identical fusants (AG/AG and HR/HR) would not grow on selection plates containing both antibiotics (Fig. 1E). Fusion frequencies determined by growth on both antibiotics are therefore an underestimate of true fusion rates. Briefly, cultures of AG and HR were first washed to remove selective antibiotics present in the medium and subsequently resuspended in a buffer containing DNase I to avoid DNA uptake from dead cell matter. Centrifuging mixtures of AG and HR at 500 $\times$ *g* resulted in the highest fusion efficiency (1.5 in 10$^5$ cells); however, the pellet formed in this case was difficult to handle. Increasing centrifugation to 1,000 $\times$ *g* reduced the fusion efficiency to less than 1 fused cell per 10$^5$ cells, and no fusion was observed at speeds above 6,000 $\times$ *g*, potentially due to cell lysis (Fig. 2A). The fusion efficiency in the presence of PEG was highest at 10 w% PEG, with 1 fused cell per 10$^5$ cells (Fig. 2B). Higher PEG concentrations, such as 50 w%, which is commonly used for protoplast fusion, caused dramatic cell lysis,

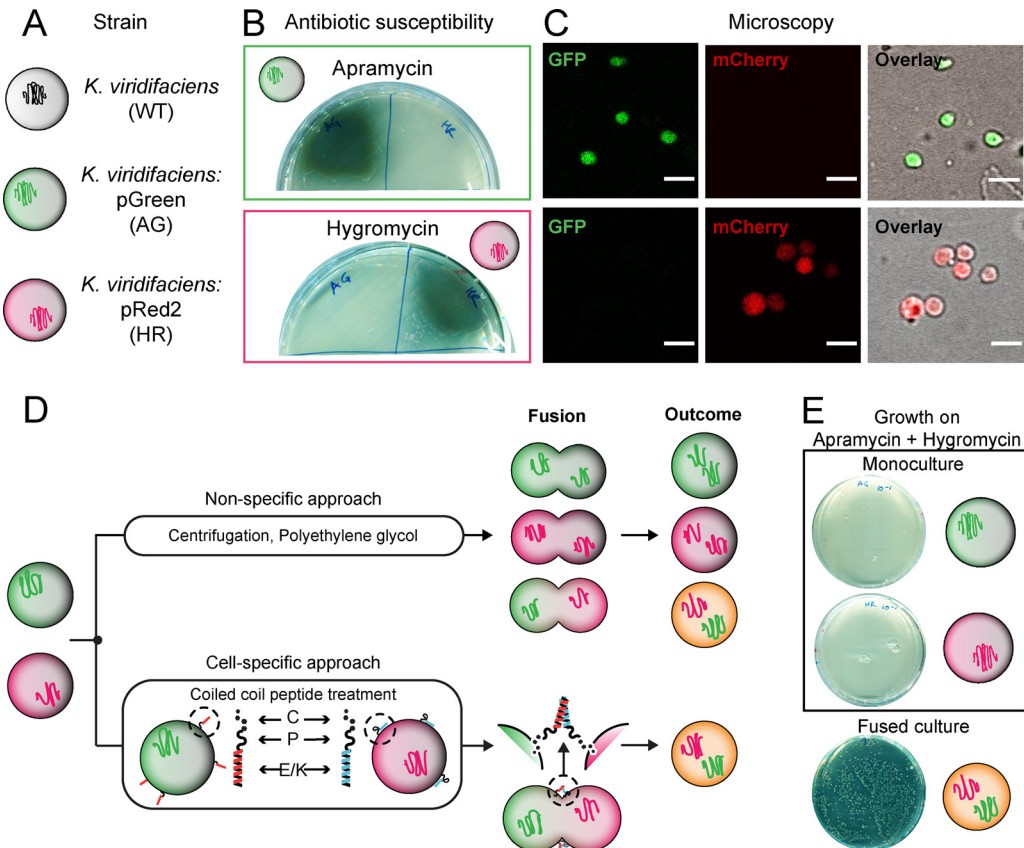

**FIG 1** L-form system and cell-cell fusion. (A) The wild-type *Kitasatospora viridifaciens* delta L-form strain was genetically modified to either express apramycin resistance and green fluorescence (AG) or hygromycin resistance and red fluorescence (HR). Each reporter pair (antibiotic resistance plus fluorescence gene) was introduced via a plasmid by using the ϕC31 integration system. (B) Strains AG and HR plated on a single antibiotic (either apramycin or hygromycin) to confirm resistance. AG showed growth on apramycin (top) and HR showed growth on hygromycin (bottom), with no cross-resistance observed. (C) Visual confirmation of fluorescence reporters by microscopy indicated a positive signal in the green channel for AG and in the red channel for HR. No bleed-through was observed between channels. Bar, 5 $\mu$m. (D) Fusion can be obtained by two approaches, namely, nonspecific (via centrifugation or PEG) and cell specific (coiled-coil lipopeptides). The process of fusion and the outcome differs in both cases. For nonspecific fusion, the membranes come together by dehydration induced by PEG or by physical centrifugal force. In case of coiled-coil lipopeptides (CPE$_4$ and CPK$_4$), they dock in the membrane using the cholesterol anchor and pull together opposing membranes upon complementary coiling. This complementarity results in fusion of only oppositely labeled cells, unlike that in the nonspecific methods. (E) Confirmation of phenotype after fusion. Monocultures of AG and HR plated on medium with both antibiotics showed no growth. Only cells that have undergone fusion grow in the presence of both apramycin and hygromycin.

suggesting that the membrane composition of L-forms is possibly different from that of protoplasts (19).

To verify that the cells growing on plates with both antibiotics (Fig. 1E) were true fusants, we performed additional growth experiments and microscopy. Different strain types were plated on four selection environments (see Fig. S1 in the supplemental material). As expected, the parental strains (AG and HR) both individually and mixed in a 1:1 ratio showed growth without antibiotic selection, as did the fusant strain derived from these parents. Furthermore, the apramycin-resistant parent (AG) alone did not grow under hygromycin selection, while the hygromycin-resistant strain (HR) did not grow under apramycin selection (see Fig. S1A). Interestingly, the mixed culture was unable to grow in the presence of both antibiotics, unlike the fusant strain (see Fig. S1B). This pattern of growth indicates that fusion of cells is a prerequisite for growth in the presence of both antibiotics. To corroborate these findings, a small patch of biomass growing on medium with both antibiotics as well as a liquid culture grown from this biomass were imaged using fluorescence microscopy (Fig. 2C). The percentage of

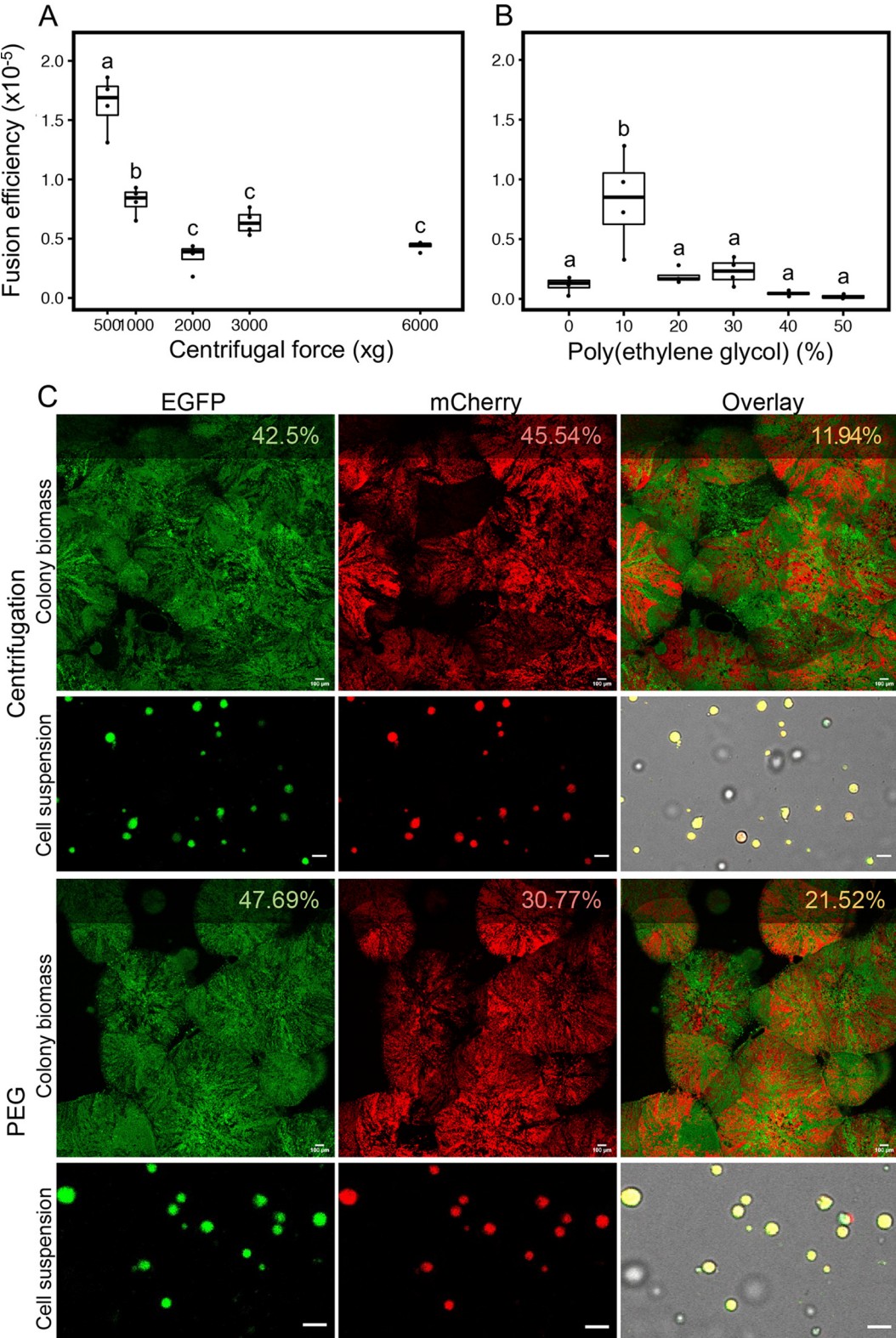

**FIG 2** Cell-cell fusion of L-forms. (A and B) Nonspecific cell fusion was carried out using either a physical method (centrifugation) (A) or chemical method (PEG) (B). The fusion efficiency was calculated by dividing the total cell count obtained on double selection media with the cell count of the individual parent strain (AG or HR). Increasing centrifugal force leads to a decrease in efficiency (one-way analysis or variance [ANOVA], $F = 15$, $P = 9.77 \times 10^{-9}$, group-wise comparison with Tukey's honestly significant difference [HSD]). PEG concentrations also affected fusion efficiency (one-way ANOVA, $F = 22$, $P = 0.033$, group-wise comparison with Tukey's HSD) with 10 w% resulting in the highest efficiency

pixels that were double labeled (i.e., containing both green and red emission) was higher for cells that had undergone fusion via PEG (21.52%) than those produced by centrifugal force (11.94%). These patches of double-labeled cells indicated the presence and subsequent expression of both sets of markers (AG and HR). This can be more clearly seen in images of cell suspensions, where the majority of individual cells expressed both EGFP and mCherry. The presence of green and red patches in the colonies can be attributed to the fact that the polyploid L-forms may consist of an unequal ratio of the two chromosome types. An unequal ratio and expression of markers can lead to a predominantly green (more AG than HR) or red (more HR than AG) colony appearance. Taken together, these results show that cell-cell fusion of L-forms is possible and that the resulting colonies are potentially heterokaryons containing all markers introduced on the two chromosomes.

**Fused cells are viable and proliferate and produce more antibiotics than nonfused cells.** Successful cell-cell fusion events between different L-form strains combine the cytoplasmic contents and genomes of these cells. To study whether these fused cells (i.e., fusants) are viable, time-lapse microscopy of individual cells was performed. In viable growing L-forms, membrane extension and blebbing take place first, along with deformation of cell shape (16, 20). This is followed by daughter cell formation, as the daughter cell tends to remain attached to the mother cell. Given the nonbinary nature of cell division in wall-deficient cells, it was difficult to track the exact number of daughter cells originating from one mother cell. Using the wild-type L-forms as a reference for cell growth, we looked for the same pattern in fused cells which were viable in the presence of both antibiotics. Colonies from a fusion event were inoculated in liquid medium with both antibiotics to obtain suspended cultures that could be introduced into a 96-well plate for time-lapse imaging in an automated microscope. We applied bright-field and fluorescence imaging every 10 min over a period of 16 h (Fig. 3; see also Video S1 in the supplemental material). Importantly, the fused L-forms followed the growth characteristics of wild-type or parental strains, as evidenced by blebbing and membrane deformation, as well as smaller daughter cells that were visibly attached to mother cells (Fig. 3; see also Videos S1 and S2). Lysed cells, on the other hand, immediately lost fluorescence and shape (see Video S3). The fusants also showed growth upon subculture into fresh medium containing both selection pressures (see Fig. S2A). These cultures were further subjected to PCR-based genotyping to confirm the presence of both marker sets (see Fig. S2B). As expected, amplification of fragments corresponding to the apramycin and hygromycin resistance genes was observed, confirming the presence of both markers. Additionally, to confirm that fusant colonies were not formed by DNA uptake and recombination of markers, individual colonies were restreaked on media without any antibiotics. The biomass was subjected to fluorescence microscopy after 3 days of growth (see Fig. S2C). Distinct regions of only EGFP-expressing or only mCherry-expressing cells were observed, indicating segregation of the different chromosomes in daughter cell populations. Colocalization analysis of pixels in these images showed a negative correlation between the two channels, further confirming segregation.

The parental *K. viridifaciens* strain, as well as L-forms, are known to produce antimicrobials such as tetracycline (14, 21). We asked if this activity is differently expressed in the fusants, since these cells were selected to maintain a minimum set of two distinct chromosomes that are essential to survive selective plating. Such a selection pressure would result in a population of cells with greater ploidy, since more chromosomes would mean greater probability of having one of each type. Fusants and the parent strains were grown in liquid culture and tested for their ability to inhibit *Escherichia coli*

**FIG 2** Legend (Continued)

of fusion. (C) Fluorescence microscopy of colony biomass on double antibiotic media after fusion via centrifugation (top two rows) or via PEG 10 w% (bottom two rows). Fluorescence expression (EGFP and mCherry) is indicated as a percentage in the top right corner of each image and was calculated using ImageJ/Fiji. The overlay image (third column) shows the percentage or area occupied by both green and red pixels and was slightly higher for PEG-induced fusion. A cell suspension from colony biomass that was cultured in LPB medium with both antibiotics and imaged at a higher magnification also shown. Scale bar for colony biomass = 100 $\mu$m; scale bar for cell suspension = 5 $\mu$m.

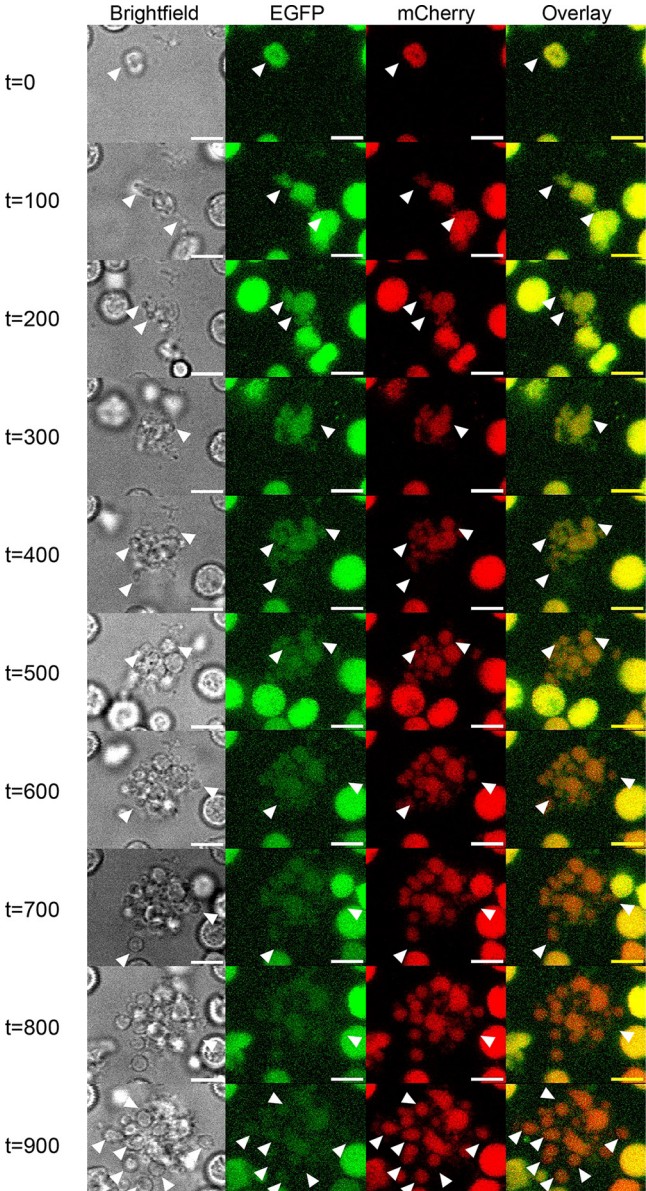

**FIG 3** Viability of fused cells. Growth and division of fused cell were tracked over time with bright-field (BF) and fluorescence (GFP and mCherry) microscopy. Images were taken every 10 min for a total of 16 h. The panels (column-wise, left to right: bright-field, EGFP, mCherry, overlay) consist of a select few images over this time period (labeled on the left, in minutes). White triangles indicate growing cells and membrane extensions, while arrows indicate a lysed cell. Cell growth was characterized by deviation from the circular shape, membrane extension, and formation of smaller circular daughter cells. Fluorescence was maintained during this process of cell growth, as seen in the EGFP and mCherry channels. Lysed cells, on the other hand, immediately lost fluorescence and shape ($t = 800$ and $t = 900$). Images were taken from Video S1, available in the supplemental material.

JM109 during coculture in a Transwell plate. The effect of each strain on the indicator strain was quantified via cell density (see Fig. S3A) and CFU counts (see Fig. S3B) over 48 h. Surprisingly, the fusants showed significantly higher antimicrobial activity than the monoculture of the indicator strain in the same growth environment (Dunnett's pairwise comparison, $P = 0.0083$; $n = 8$). We also observed a significant difference between the parent and fusant strains in the CFU of *E. coli* after coculture, indicating that fused cells could produce increased concentrations of inhibitory compounds (Dunnett's pairwise comparison, $P = 0.0033$; $n = 8$). Our results demonstrated that

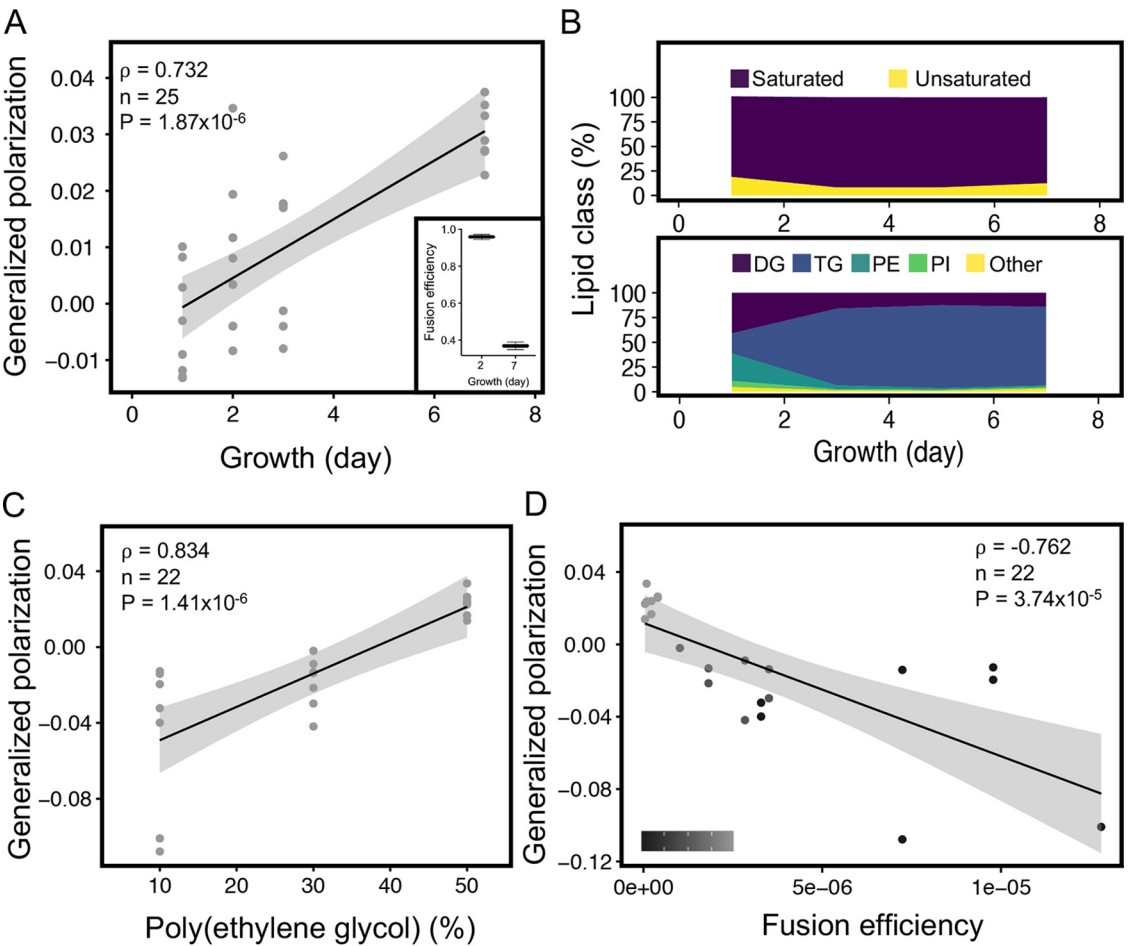

**FIG 4** Membrane fluidity affects L-form fusion. (A) Fluidity of L-form membranes was quantified as a generalized polarization (GP) value, using the Laurdan dye assay. A strong positive correlation was obtained between GP value and the period of growth, indicating a decrease in membrane fluidity with increasing culture age (Spearman's rank correlation test). Age of the culture also had an effect on fusion efficiency (inset, 2-sample $t$ test, $P = 2.22 \times 10^{-6}$, $n = 3$) with young 2-day-old cultures fusing more efficiently than older 7-day-old cultures. (B) Analysis of membrane lipids of cultures from different periods of growth (1, 3, 5, and 7 days) indicated a change in the percentage of saturated and unsaturated fatty acids over time. Specifically, triglyceraldehyde (TG) and phosphatidylethanolamine (PE) showed a strong decrease between days 1 and 3. Both lipids are required for fluidity of the membrane. (C) Positive correlation obtained between GP value and the percentage of PEG used, indicating a decrease in membrane fluidity with increasing concentration of PEG (Spearman's rank correlation test). (D) The GP value showed a strong negative correlation with fusion efficiency. A low percentage of PEG (10%) leads to slightly more fluid membranes compared to those exposed to a high PEG percentage (50%), which resulted in higher fusion (Spearman's rank correlation test). The grayscale (bottom left corner) indicates the PEG percentage, which ranged from 10 to 50%.

**Membrane fluidity influences fusion efficiency.** The bacterial cell membrane largely consists of (phospho)lipids and fatty acids, together with other minor components. The characteristics of these lipids and fatty acids (FAs), such as the degree of unsaturation and headgroup composition, determine the physical properties of a membrane. The fluidity of a membrane is an important factor governing its fission and fusion abilities (22, 23). Membrane fluidity of L-form cells was quantified as generalized polarization (GP) using the Laurdan dye assay (24). This GP value can range from $-1$ to $+1$ and inversely correlates to membrane fluidity (i.e., a low GP value indicates a more fluid membrane). Measuring the fluidity for L-forms grown for 1, 3, 5, and 7 days resulted in a significant GP value increase over time (Fig. 4A) (rho = 0.732, $P = 1.87 \times 10^{-6}$), indicating that membrane rigidity increases as the cultures age. Importantly, this change in fluidity with culture age negatively correlated with the fusion efficiency, as younger cultures fused at twice the efficiency

of older cultures (Fig. 4A, inset) (unpaired $t$ test, $P = 2.22 \times 10^{-6}$). To assess the underlying molecular causes for this shift in fluidity, the membrane lipid and FA composition were analyzed using mass spectrometry (MS) for L-form cultures of different ages. Over a 7-day period, there was a significant shift in (phospho)lipid and FA composition as the fraction of saturated FAs increased at the expense of unsaturated FAs (Fig. 4B, top panel). This change is consistent with previous reports for *Streptomyces* spp. and *Bacillus* spp. showing that membrane fluidity decreases due to the presence of saturated FAs that stack tightly and thereby make membranes rigid (19, 22). In addition, the percentage of phosphatidylethanolamine (PE), which is known to affect membrane curvature, declines with culture age in L-forms. Both factors, an increase in saturated FAs and a decrease in PE, likely underlie the shift in fusion frequency with colony age, although by different mechanisms.

To confirm the impact of membrane fluidity on fusion efficiency, we directly manipulated membrane fluidity by adding PEG into the medium, which is known to induce fusion between two membranes by hydrogen bonding and to force adjacent membranes into close proximity via dehydration (8, 10). When we tested the effect of increasing PEG concentrations on L-form membrane fluidity, we observed a significant positive correlation between GP values and PEG concentrations (rho = 0.834, $P = 1.41 \times 10^{-6}$) (Fig. 4C). This showed that an increase in PEG leads to reduced membrane fluidity in L-forms. In turn, this causes a decrease in fusion efficiency. Thus, a high GP value (i.e., low membrane fluidity) results in low fusion (rho = $-0.762$, $P = 3.74 \times 10^{-5}$) (Fig. 4D).

Taken together, these results show that increased membrane fluidity facilitates fusion, which varies naturally during the growth of L-form cells and can be chemically manipulated by the addition of PEG.

**Coiled-coil lipopeptides localize to L-form membranes and alter membrane fluidity.** PEG-mediated fusion and centrifugation cause nonspecific cell fusion, and this can result in a low percentage of fused cells expressing both EGFP and mCherry (Fig. 1D). The recent use of lipidated peptides in cell fusion has shown great promise to improve fusion efficiency, with examples of successful fusion between liposomes or liposomes with various eukaryotic cell lines (25–28). A coiled-coil is a common protein structural motif (see Fig. S4) that contains two or more alpha-helices wrapped around each other to form a left-handed superhelical structure (29, 30). In previous studies, *de novo*-designed coiled-coils forming lipopeptides $K_4$ and $E_4$ were conjugated to cholesterol via a flexible PEG-4 spacer, yielding lipopeptides denoted $CPK_4$ and $CPE_4$ (31, 32). Using this coiled-coil membrane fusion system, efficient liposome-liposome and cell-liposome fusion has been achieved, resulting in efficient cytosolic delivery of cargo (25–27). Since L-forms do not possess a cell wall and the outer membrane is structurally similar to (giant) lipid vesicles, we investigated whether coiled-coil lipopeptides $CPE_4$ and $CPK_4$ can be applied to increase the L-form fusion efficiency and introduce cell specificity. First, we tested whether lipopeptide $CPK_4$ could be inserted in the L-form membrane and still form a coiled-coil with its binding partner, peptide $E_4$ (Fig. 1D; see also Fig. S4). Incorporating the $CPK_4$ lipopeptide in the membrane allowed docking of the complementary fluorescent-labeled peptide $E_4$ (fluo-$E_4$) (Fig. 5A). Docking was also observed when $CPE_4$ was incorporated in the L-form membrane followed by the addition of fluorescent-labeled peptide fluo-$K_4$. In contrast, no fluorescence was observed when only fluo-$K_4$ or fluo-$E_4$ was added to L-forms (Fig. 5B). Using image analysis software, we further confirmed membrane localization of the lipopeptide-fluorescent dye conjugate by assessing the fluorescence intensity across the cell along a transect line. A combined plot (see Fig. S5) of these intensity values across 10 cells indicated coinciding peaks of fluorescence values of the lipopeptide conjugates with that of values for the cell membrane (seen as dark gray rings in bright-field images). The fluorescence intensity on L-form membranes was more distinct when $CPE_4$/fluo-$K_4$ was used, compared to $CPK_4$/fluo-$E_4$ (see Fig. S5A). Altogether, these results demonstrate for the first time that lipopeptides can be readily incorporated into L-form membranes and serve as a docking point for the complementary (lipo)peptides.

The incorporation of lipopeptides in L-form membranes prompted us to investigate whether they also influenced membrane fluidity. To test this, L-forms expressing red fluorescent protein (AR and HR strains) were modified with either non-fluorescent-labeled

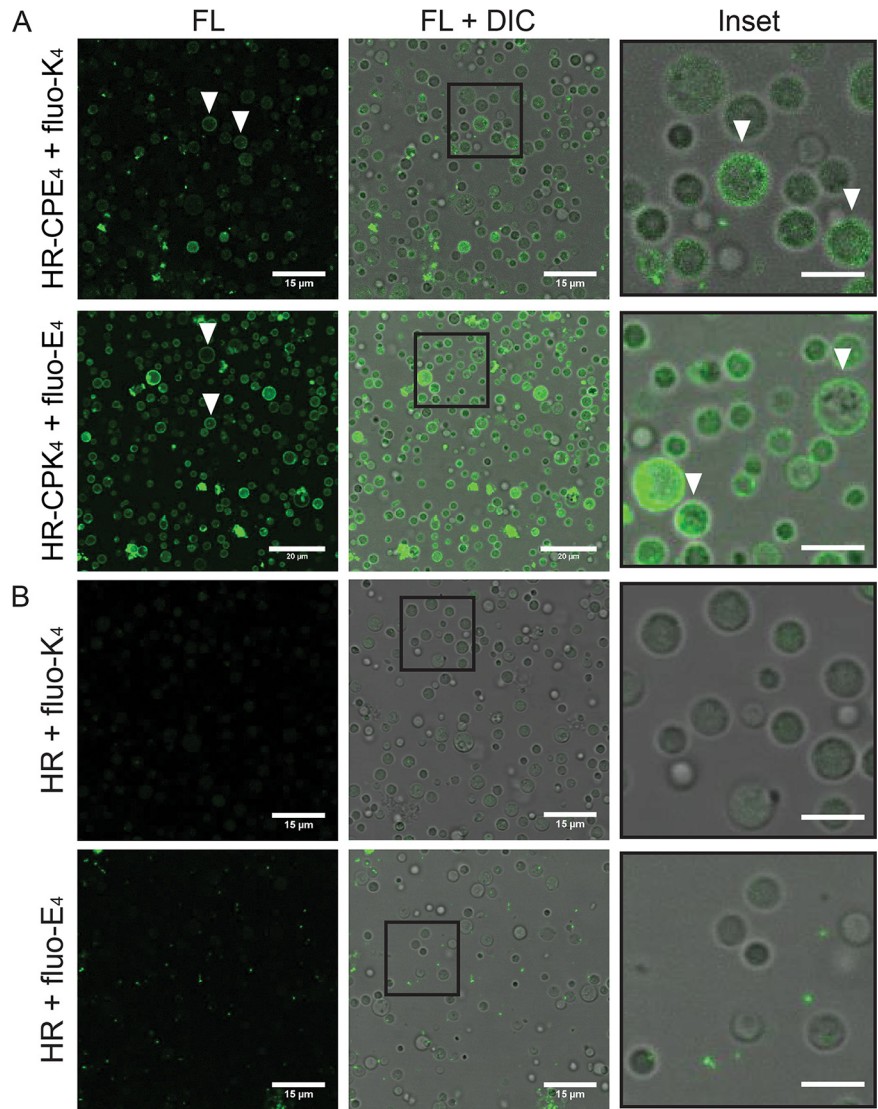

**FIG 5** Coiled-coil lipopeptides integrate in L-form membranes. (A) Confocal microscopy images (fluorescence [FL] and overlay [FL +differential interference contrast, or DIC]), indicating peptide $CPE_4$ or $CPK_4$ insertion into the L-form membranes and coiled-coil formation with complementary peptides (fluo-$K_4$ or fluo-$E_4$). White arrows indicate clear membrane insertions. (B) In the absence of $CPE_4$ or $CPK_4$, no binding of the complementary fluorescent peptide (fluo-$K_4$ or fluo-$E_4$) was observed. Experiments were performed at 30°C, and L-forms in P-buffer were incubated with 10 $\mu$M $CPE_4$ or $CPK_4$ for 30 min. Subsequently, the unbound peptide was washed via centrifugation and the complementary fluorescent peptides were added. Scale bar = 5 $\mu$m.

$CPE_4$ or $CPK_4$, so as not to interfere with the emission spectra of the Laurdan dye. The observed GP values revealed that $CPK_4$ and $CPE_4$ affect the fluidity of L-forms differently. While $CPE_4$ decreased fluidity in the AR strain (Fig. 6A), both lipopeptides increased fluidity in the HR strain (Fig. 6B). Interestingly, the effect of increased fluidity due to PEG (10% wt/wt) was only observed in the AR strain. These differences in fluidity effects are likely caused by the presence of antibiotics during culturing of the strains prior to the experiment, which is required to avoid contamination in the cultures (see Fig. S6). Antibiotics are known to affect membrane fluidity (33); however, the exact mechanism by which they do so is unclear. This inherent difference was observed in the basal GP values of control samples (−0.02 for HR and −0.08 for AR) (Fig. 6A and B), as well as in separate measurements of fluidity for strains in the absence and presence of antibiotics (0.01 for HR and −0.10 for AR) (see Fig. S6). However, all treatments (PEG and lipopeptide) were

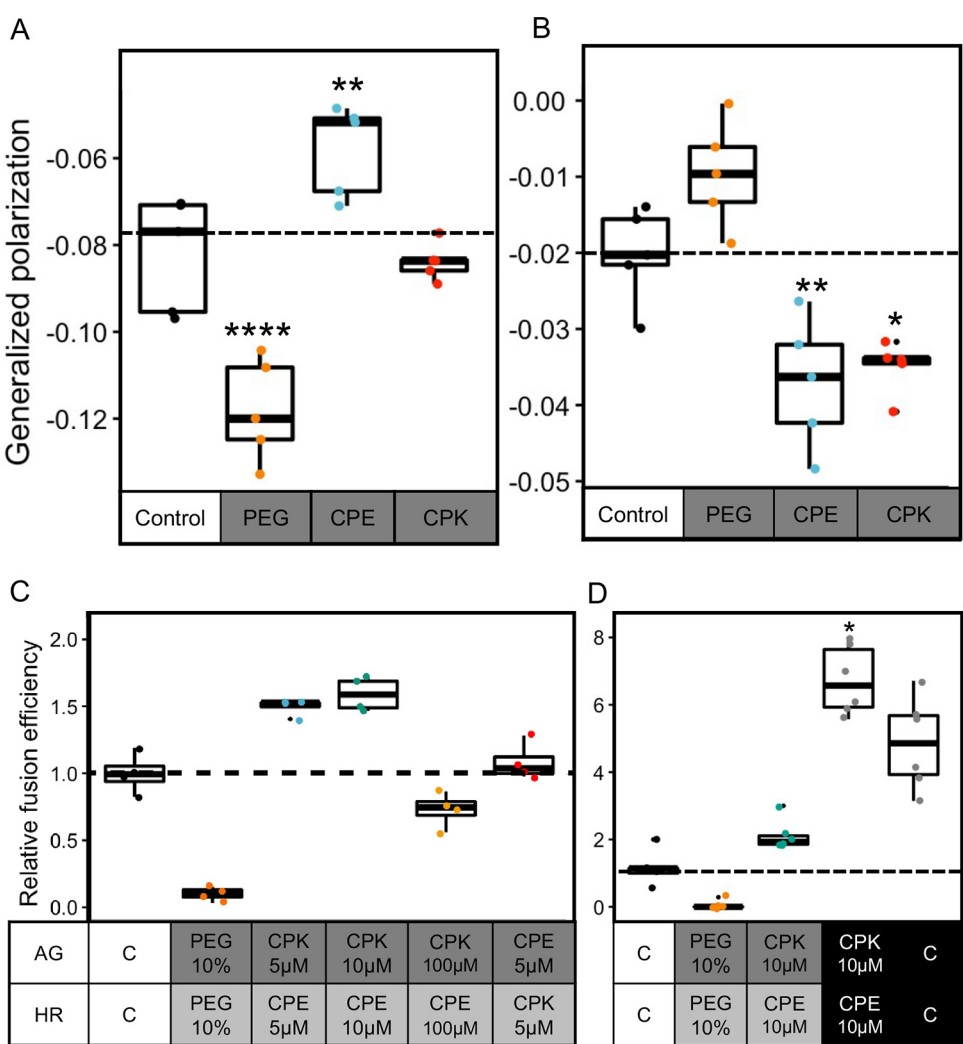

**FIG 6** Coiled-coil lipopeptides increase membrane fluidity and cell-specific fusion. (A) Strain AR showed an increased fluidity on treatment with PEG ($P = 3.06 \times 10^{-6}$), a decrease in fluidity on treatment with $CPE_4$ ($P = 2.13 \times 10^{-3}$), and no change in fluidity with $CPK_4$ (one-way ANOVA, $F = 36$, $P = 4.59 \times 10^{-18}$, followed by Tukey's pairwise comparison) compared to the control (dotted line). (B) Strain HR showed increased fluidity (low GP value) when treated with $CPE_4$ ($P = 3.11 \times 10^{-3}$) and $CPK_4$ ($P = 1.4 \times 10^{-2}$) compared to the control (dotted line), whereas no significant change was found when treated with 10% PEG (one-way ANOVA, $F = 36$, $P = 2.83 \times 10^{-18}$, followed by Tukey's pairwise comparison). Dotted line is for comparison of GP values to the control where no peptide or PEG was added. (C) The AG and HR strains were individually treated with either PEG, $CPE_4$, or $CPK_4$ at different peptide concentrations to assess the effects on fusion efficiency. Interestingly, PEG resulted in a low level of fusion despite increasing fluidity, because of its nonspecific nature. The combination of AG-$CPK_4$ and HR-$CPE_4$ resulted in the highest fusion efficiency relative to the basal level. The increase in relative fusion efficiency was concentration dependent as well as peptide dependent (one-way ANOVA, $F = 30$, $P = 3.47 \times 10^{-14}$, followed by Tukey's pairwise comparison). (D) The AG and HR strains were first treated with either PEG, $CPE_4$, or $CPK_4$. These strains were then directly plated on double selection media in the absence (gray boxes) or presence (black boxes) of 10 w% PEG to assess the effect on fusion efficiency. Interestingly, PEG resulted in low fusion, despite increasing fluidity because of its nonspecific nature when washed away prior to plating, but it produced a high efficiency when present during the plating. The treatment with peptides also showed a higher efficiency when in the presence of PEG (Kruskall-Wallis chi-square = 24.84, $P = 5.4 \times 10^{-5}$, followed by Dunnett's pairwise comparison) compared to the control, where no peptide or PEG was added (dotted line).

compared to the control sample of the individual strain type; hence, the change in GP value was indeed due to the lipopeptide interaction and not due to the presence of antibiotics.

We next examined how these changes in fluidity affect the process of lipopeptide-mediated fusion. For this, L-form cultures were first adjusted to the same density and split into aliquots. The aliquots were then either untreated (control), treated with PEG,

or treated with increasing concentrations of the lipopeptide that previously caused an increase in fluidity. HR strains were hence pretreated with $CPE_4$ and AG strains were treated with $CPK_4$. After treatment for 30 min, the excess PEG and lipopeptides were removed by centrifugation and the L-forms were resuspended in fresh P-buffer containing DNase I. The cultures were then thoroughly mixed in a 1:1 ratio, incubated for 30 min at 30°C, and subsequently plated on selection media for cell quantification. The observed fusion efficiency for each treatment relative to control revealed that treatment of HR with $CPE_4$ and of AG with $CPK_4$ resulted in a high fusion efficiency compared to that with 10 w% PEG or the centrifuged control (Fig. 6C). Furthermore, fusion efficiency was not only dependent on lipopeptide concentration (i.e., decreased fusion at 100 $\mu$M) but also on the lipopeptide specificity, since AG treated with $CPE_4$ resulted in a basal level of fusion, i.e., similar to the control. Higher lipopeptide concentrations also visibly affected cells, causing lysis (data not shown). Together, these results confirm that cell-specific fusion of L-forms can be achieved using fusogenic coiled-coil lipopeptides.

The two approaches (nonspecific via PEG or centrifugation and cell-specific using lipopeptides) employed here seem to influence fusion by altering membrane fluidity and bringing membranes together. We then investigated whether combining both fusogens would result in an overall higher fusion efficiency. For this, the cells were first treated with the lipopeptides (AG L-forms with $CPK_4$ and HR L-forms with $CPE_4$) and split into two aliquots. The first aliquot was directly subjected to fusion by mixing the cultures in a 1:1 ratio, whereas the second aliquot was mixed and treated with 10% PEG. Here, the PEG remained in the environment during the process of fusion. Efficiency calculations showed 3-fold-higher relative fusion in the latter (Fig. 6D), indicating that combining lipopeptides and PEG is optimal for cell-cell fusion. The presence of lipopeptides on the cell surface aids in complementary L-form pairing (AG with HR), bringing the opposing membranes in proximity, which is an important first step in fusion. Additionally, PEG potentially further reduces the space by membrane dehydration, thus facilitating fusion events. Colony imaging further confirmed the presence of more double-labeled cells after treatment with PEG (see Fig. S7).

## DISCUSSION

Cell wall deficiency has primarily been studied in the context of stress tolerance and intracellular pathogenicity (34). The genetic and metabolic modifications required to survive in a wall-deficient state are also being uncovered, which has deepened our understanding of their intriguing biology (15, 35). We have shown that wall-deficient L-forms can fuse with one another and that membrane fluidity is a key factor influencing fusion efficiency. Additionally, we have shown for the first time targeted fusion between wall-deficient cells using coiled-coil lipopeptides. This opens avenues for applications in the field of biotechnology and the design of synthetic cells.

Successful fusion between cells should result in mixing of cytoplasmic contents. In this study, the traceable contents (antibiotic resistance and fluorescence proteins) were mixed. Time-lapse microscopy along with PCR-based quantification of the fused population in media with both antibiotics indicated that at least one copy of each chromosome was present in the cell. Given that these L-forms are often polyploid due to the absence of regulated cell division, there is a possibility of an unequal ratio of both chromosomes (16). This imbalance can then lead to differences in the expression of markers and other genes, such as those for antibiotic biosynthesis. In the presence of antibiotic selection pressure, the fused population will likely consist of a higher proportion of polyploid cells compared to the ancestral strains. This could explain why the ancestral strains show greater variability when it comes to antimicrobial activity against indicator strains. The ancestral population, due to an absence of selection pressure, can consist of monoploid cells too, which would affect overall antimicrobial activity. This variability also indicates a potential influence of ploidy on the antibiotic production capacity of a cell.

L-forms are surrounded by a membrane which is sufficiently fluid to allow efficient proliferation. *Bacillus subtilis* L-forms that have a defect in formation of branched-chain fatty acids (BCFAs) suffer from decreased membrane fluidity and, as a consequence, cannot carry out the membrane scission step (22). This phenotype was rescued by supplementing the medium with BCFAs in the medium. Less is known about the impact of fluidity on bacterial fusion, although older reports on eukaryotic muscle cell cultures suggested that myoblast fusion is preceded by a decrease in membrane viscosity (23). In this work, we showed that the membrane fluidity of *K. viridifaciens* L-forms changes over time. In younger cultures, the fluidity is higher, coinciding with the ability of such cells to proliferate efficiently. By contrast, the fluidity decreases in older cultures. The change in fluidity is associated with a change in the ratio of saturated to unsaturated FAs. In our study, we found this ratio to be 4.3 for the first day of growth, which then increased to 11.3 after 3 days (Fig. 4B). Thus, the amount of saturated FAs responsible for tighter packing increases over time at the expense of unsaturated FAs. The accumulation of saturated FAs makes the membrane more stiff, which negatively impacts proliferation and fusion efficiency. Notably, compared to protoplasts, L-forms of *Streptomyces hygroscopicus* contain 6 times more anteiso FAs than do protoplasts, resulting in more fluid membranes (19). Our lipidomic analysis also indicated that L-form membrane composition comprised significant amounts of cardiolipin (CL), phosphatidylinositol (PI), and phosphatidylethanolamine (PE). Both CL and PE are fusogenic headgroups shown to induce fusion between liposomes and extracellular vesicles (36), and their presence may also facilitate L-form fusion.

A pair of complementary fusogenic coiled-coil lipopeptides were previously developed for the targeted delivery of compounds into eukaryotic cells via liposomes. These eukaryotic liposome models have also been used extensively to better understand the process of cell fusion (37). For the first time, we explored here targeted fusion with these synthetic lipopeptides between bacterial cells. Interestingly, we observed that the lipopeptides readily inserted in membranes of L-forms via a cholesterol anchor (Fig. 5). These lipopeptides remained in the membrane even after several washing steps. The lipopeptide segment of $CPK_4$ is known to interact both with its binding partner peptide $E_4$ as well as membranes, while the peptide $E_4$ segment of $CPE_4$ does not (Fig. 1D). Complementary binding of the lipopeptides brings two opposing membranes into close proximity and ultimately induces fusion (29, 38). The differences in lipopeptide presentation on the surface can explain the complementarity effect on fusion efficiency of L-forms as well (Fig. 6). Given the ease of lipopeptide docking and subsequent stability on the L-forms, coiled-coil lipopeptides provide a promising avenue for studies on targeted compound delivery into wall-deficient cells. This may be particularly relevant for L-forms associated with recurring urinary tract infections and potentially mycobacterial infections (39, 40).

The costs and benefits of living as a wall-deficient cell depend on the environment. Absence of a protective wall makes them sensitive to changes in osmotic pressure and physical agitation. On the other hand, cells without a wall are resistant to a whole class of cell wall-targeting antibiotics (penicillins, cephalosporins), transport to the extracellular space is potentially easier, and the cells are stably polyploid. These characteristics can make L-forms a unique model system to study not only cell biology but also questions in the fields of biotechnology, evolution, and the origin of life (34, 41, 42). First, our use of coiled-coil-directed fusion can be extended to synthetic cells to obtain fusions that increase cellular complexity as well as interspecies fusions. Second, fusion leads to multiple chromosomes in the same cellular compartment, which in turn can result in genetic recombination. Such recombination events can then be leveraged to identify new microbial products and obtain genomically diverse populations of cells. Finally, cell-cell fusion can also help increase understanding of major transitions on the road to increased organismal complexity, like multicellularity and endosymbiosis.

## MATERIALS AND METHODS

**Media and growth conditions.** All L-form strains were cultured in liquid L phase broth (LPB) and solid L phase medium agar (LPMA). LPB consists of a 1:1 mixture of yeast extract malt extract (YEME) and

tryptic soy broth supplemented with 10% sucrose (TSBS) and 25 mM MgCl$_2$. LPMA consists of LPB supplemented with 1.5% agar, 5% horse serum, and 25 mM MgCl$_2$ (9). P-buffer containing sucrose, K$_2$SO$_4$, MgCl$_2$, trace elements, KH$_2$PO$_4$, CaCl$_2$, and N-tris (hydroxymethyl)-methyl-2-aminoethanesulfonic acid (TES) (9) was used for transformation and all fusion experiments and was supplemented with 1 mg/mL DNase I (Roche Diagnostics GmbH). The antibiotics apramycin (Duchefa Biochemie) and hygromycin (Duchefa Biochemie) were used for selection and were added at final concentrations of 50 $\mu$g/mL and 100 $\mu$g/mL, respectively. Growth conditions for all cultures were 30°C in an orbital shaker (New Brunswick Scientific Innova) with 100 rpm for the liquid cultures. Centrifugation (Eppendorf centrifuge 5424) conditions were always 1,000 $\times$ $g$ for 10 min (<1 mL) or 30 min (>10 mL), depending on culture volume. The above-mentioned culture conditions and centrifugation settings were applied throughout the study unless mentioned otherwise. All measurements for optical densities (ODs) of samples were done with 200-$\mu$L aliquots of culture in a 96-well flat-bottom plate (Sarstedt) using the Tecan Spectramax plate reader.

**Strain and plasmid construction.** Wall-deficient L-forms of *Kitasatospora viridifaciens* were obtained by prolonged exposure to penicillin and lysozyme, similar to a previous study (43). Briefly, 10$^6$ spores of *Kitasatopsora viridifaciens* DSM40239 were grown in 50 mL TSBS medium at 30°C and 100 rpm to obtain mycelial biomass. To this biomass, 1 mg/mL lysozyme (Sigma-Aldrich) and 0.6 mg/mL penicillin (Duchefa Biochemie) were added to induce S-cell formation. After 7 days, a dense culture of wall-deficient cells was obtained and subcultured to LPB medium containing 6 mg/mL penicillin. This treatment was continued for 5 weeks with subculture into fresh medium every week. The culture was then tested for growth on LPMA without penicillin, and it showed only L-form growth. A single colony was picked and inoculated in LPB without penicillin and incubated for 7 days to confirm stability of the wall deficiency and subsequently used for making a culture stock to be stored at −80°C.

The strain was further genetically modified to harbor antibiotic resistance genes and fluorescent reporter genes. Two plasmids were used for this purpose namely, pGreen [containing the apramycin resistance gene *aac(3)IV* and a green fluorescent protein reporter gene] and pRed2 (containing the hygromycin resistance gene *hph* and a red fluorescent reporter gene). Both plasmids contain the ϕC31 *aatP* site and a ϕC31 integrase, which allows for integration of the marker set at the *attB* site in the genome. The pGreen plasmid was obtained from a previous publication where details can be found regarding the construction (44). The pRed2 plasmid was constructed by introducing the amplified mCherry gene along with a *gap1* promoter region at the XbaI site in the pIJ82 plasmid. Briefly, the mCherry gene was amplified together with the *gap1* promoter using primers (Sigma) mentioned in Table S1 and with the pRed plasmid (44) as template. The amplified gap1-mCherry product was purified using a kit following the instructions of the supplier (Illustra GFX gel band purification kit). The purified product was introduced into the vector pIJ82 at the XbaI site (New England Biolabs GmbH). This plasmid was first transformed into *E. coli* DH5$\alpha$ for amplification followed by transformation into *E. coli* ET12567 for demethylation.

The plasmids were introduced into the L-forms by PEG-induced transformation, a method similar to that for protoplast transformation with some modifications (9). L-form cultures were grown for 4 days. Cultures were centrifuged to remove the spent medium, and the pellet was resuspended in a 1/4 volume of P-buffer. Approximately 500 ng of plasmid was added to the resuspended pellet and mixed thoroughly. PEG1000 was added to this mix at a final concentration of 25 w% and mixed gently. After a brief incubation of 5 min on the bench, the tube was centrifuged. The supernatant was discarded, the pellet was resuspended in LPB medium, and this mixture was incubated for 2 h. The culture was then centrifuged again and the pellet resuspended in 100 $\mu$L LPB for plating on LPMA containing the selective antibiotics apramycin or hygromycin. After 4 days of incubation, single colonies were picked and restreaked on LPMA with antibiotics for confirmation, along with fluorescence microscopy. The resulting strains were named AG for apramycin-green and HR for hygromycin-red.

To test the antibiotic susceptibility, both strains were grown on LPMA with or without either 50 $\mu$g/mL apramycin or 100 $\mu$g/mL hygromycin for 4 days. Stepwise 10-fold dilution plating was done, which allowed for quantifying the number of colonies (in CFU per milliliter).

**L-form fusion.** Strains AG and HR were grown individually from culture stocks in 20 mL LPB containing the relevant antibiotic. Grown cultures were then centrifuged to remove spent medium containing antibiotic and washed with P-buffer twice. The pellet was finally resuspended in 2 to 3 mL of P-buffer containing DNase I (1 mg/mL), and the density was adjusted to an optical density at 600 nm (OD$_{600}$) of 0.6. Both strains were then mixed in equal volumes (200 $\mu$L) in a fresh microcentrifuge tube and mixed gently, followed by incubation at room temperature for 10 min. Depending on the treatment, PEG1000 was added at the desired concentration (0 to 50% [wt/wt]) and mixed by pipetting. For the effect of centrifugation on L-form fusion, no PEG was added. After a brief incubation of 5 min, the tubes were centrifuged and the supernatant was discarded. The pellet was resuspended in 100 $\mu$L of P-buffer with DNase I, and serial dilutions were subsequently plated on LPMA with both antibiotics. Controls were also plated on the same medium and included 100 $\mu$L monocultures of each strain to test for cross-resistance and 100 $\mu$L of a 1:1 mix of each strain without fusion (see Fig. S1 in the supplemental material). All plates were incubated for 3 days, after which CFU were calculated to determine the fusion efficiency. Efficiency was quantified as the CFU per milliliter on double-antibiotic selection medium and normalized against the CFU per milliliter counts for monocultures grown on single-antibiotic selection medium.

**Genomic DNA isolation.** Strains were inoculated in LPB medium (20 mL) with the required antibiotic(s) for selection. The cultures were grown for 3 days, followed by centrifugation (1,000 $\times$ $g$, 30 min). Pellets were gently resuspended in a 1/10 volume of 10.3% sucrose solution (2 mL). To this suspension, 20 $\mu$L of 0.5 M EDTA (pH 8) and 400 $\mu$L of 10% SDS was added. After gently mixing, 2 mL of a phenol-chloroform-isoamyl alcohol mix (Acros Organics) and 1.5 mL of 5 M sodium chloride was added. The

tubes were mixed gently for 10 min, followed by centrifugation at 4,000 rpm for 30 min. The aqueous top layer (1 mL) was recovered and again mixed with 1 mL of phenol-chloroform-isoamyl alcohol solution. After gentle mixing for a few minutes, centrifugation was at 4,000 rpm for 30 min. The aqueous phase (500 $\mu$L) was transferred to a fresh tube, followed by addition of 500 $\mu$L of isopropanol and 50 $\mu$L of 3 M sodium acetate. DNA was allowed to precipitate overnight at $-20°C$. DNA was recovered by centrifugation at 5,000 rpm for 30 min and washed with 99% ethanol, followed by air drying the pellet. Final resuspension of the DNA pellet was done in sterile water. The DNA was quantified and diluted accordingly for the PCR mix.

**Microscopy.** A Zeiss LSM 900 Airyscan 2 microscope was used to image the fluorescently labeled strains under 40$\times$ magnification. For EGFP, an excitation wavelength of 488 nm was used and emission was captured at 535 nm, whereas for mCherry an excitation wavelength of 535 nm was used and emission was captured at 650 nm. Multichannel (fluorescence and bright-field), multistack images were captured using the Zen software (Zeiss) and further analyzed using ImageJ/Fiji. Multiple tiles were imaged for colonies to cover a large area. These tiles were then stitched, and each fluorescence channel was first subjected to thresholding to determine the total pixel area. These threshold images were then used to calculate total area (using the OR function in the Image calculator) and the fused area (using the AND function). The total area selection was then used to calculate individual pixel area occupied by either green or red pixels and by both.

The Lionheart FX automated microscope (BioTek) was used for time-lapse imaging of double-labeled L-forms after fusion. The fusant strains were precultured in LPB containing both antibiotics for 3 days. These were then centrifuged and resuspended in fresh medium with antibiotics, and 100 $\mu$L of this mixture was added to individual wells in a 96-well black, clear-bottomed Sensoplate (Thermo Fisher Scientific). The plate was centrifuged for 5 min to enable settling of cells. The time-lapse imaging was done using a 63$\times$ dry objective, set for 3 channels (bright-field, green, and red), with imaging every 10 min for 16 h at 30°C. The LED intensity for all channels was 10 with a camera gain of 24. The exposure time was set at the beginning of the imaging and was based on the reference monoculture strains AG and HR.

**Membrane fluidity assay.** The membrane fluidity was quantified for cultures of different ages and cultures treated with different lipopeptides using the Laurdan dye assay (24). All cultures grown in a 40-mL volume were first centrifuged, followed by resuspension in P-buffer, and density was adjusted to an OD$_{600}$ of 0.6 to 0.8. The cultures were then divided into aliquots according to the treatment for a given biological replicate (i.e., 5 aliquots of 1 mL each for 5 treatments). For lipopeptide treatment, the lipopeptide was added to the culture at the required concentration (5 $\mu$M, 10 $\mu$M, or 100 $\mu$M) and all tubes were incubated for 30 min at 100 rpm. Centrifugation was carried out to remove excess lipopeptide, and the pellet was resuspended in P-buffer. The P-buffer for this assay was always maintained at 30°C, so as not to alter the fluidity of the membrane. A 10 mM Laurdan (6-dodecanoyl-2-dimethylaminonapthalene, Invitrogen) stock solution was prepared in 100% dimethylformamide (DMF; Sigma) and stored at $-20°C$ in an amber tube to protect from light exposure. This stock solution was used to obtain a final concentration of 10 $\mu$M in the resuspended cultures described above. The tubes were inverted to mix the dye sufficiently and then incubated at 30°C for 10 min and covered with foil to protect from light exposure. The cultures were then washed three times in prewarmed P-buffer containing 1% dimethyl sulfoxide (DMSO; Sigma) to ensure removal of unbound dye molecules. The final suspension was performed in prewarmed P-buffer, and 200 $\mu$L was transferred to a 96-well black, clear-bottomed Sensoplate for spectroscopy. Fluorescent intensities ($I$) were measured by excitation at 350 nm and two emission wavelengths (435 and 490 nm). The background values were first subtracted from all sample values, followed by estimation of the generalized polarization (GP) value: GP = $[(I_{435} - I_{490})/(I_{435} + I_{490})]$. The GP values ranged from $-1$ to $+1$, with low values corresponding to high membrane fluidity.

**Lipid extraction and analysis.** Cultures of the wild-type L-form were grown for different time periods (1, 3, 5, and 7 days). These cultures were then centrifuged and resuspended in P-buffer prior to membrane lipidomics. Lipids were extracted using the modified methyl-tert-butyl ether (MTBE) protocol of Matyash et al. (45). In short, 600 $\mu$L MTBE and 150 $\mu$L methanol were added to the thawed bacteria samples. Samples where briefly vortexed, ultrasonicated for 10 min, and shaken at room temperature for 30 min. Next, 300 $\mu$L water was added and the samples where centrifuged for 5 min at 18,213 $\times$ g at 20°C. After centrifugation, the upper layer was collected and transferred to a glass vial. The extraction was repeated by adding 300 $\mu$L MTBE and 100 $\mu$L methanol. Samples where briefly vortexed and shaken at room temperature for 5 min. Next, 100 $\mu$L water was added and the samples were centrifuged for 5 min at 18,213 $\times$ g at 20°C. After centrifugation, the upper layer was collected, and the organic extracts were combined. Samples were dried under a gentle stream of nitrogen. After drying, samples were reconstituted in 100 $\mu$L 2-propanol. After brief vortexing and ultrasonication for 5 min, 100 $\mu$L water was added. Samples were transferred to microvial inserts for analysis.

Lipidomic analysis of bacteria lipid extracts was performed using a liquid chromatography-tandem MS (LC-MS/MS)-based lipid profiling method (46). A Shimadzu Nexera X2 (consisting of two LC30AD pumps, a SIL30AC autosampler, a CTO20AC column oven, and a CBM20A controller; Shimadzu, 's-Hertogenbosch, Netherlands) was used to deliver a gradient of water-acetonitrile starting at 80:20 (eluent A) and water–2-propanol–acetonitrile at 1:90:9 (eluent B). Both eluents contained 5 mM ammonium formate and 0.05% formic acid. The applied gradient, with a column flow of 300 $\mu$L/min, was as follows: 0 min with 40% B, 10 min with 100% B, and 12 min with 100% B. A Phenomenex Kinetex C$_{18}$ column with 2.70-$\mu$m particles, 50 by 2.1 mm (Phenomenex, Utrecht, the Netherlands) was used as the column with a Phenomenex SecurityGuard Ultra C8 cartridge, 2.7-$\mu$m particles, 5 by 2.1 mm as guard column. The column was kept at 50°C. The injection volume was 10 $\mu$L.

The MS instrument was a Sciex TripleTOF 6600 (AB Sciex Netherlands B.V., Nieuwerkerk aan den Ijssel, the Netherlands) operated in positive electrospray ionization (ESI+) and ESI− modes, with the following conditions: ion source gas 1 at 45 lb/in$^2$, ion source gas 2 at 50 lb/in$^2$, curtain gas at 35 lb/in$^2$, temperature of 350°C, acquisition range of $m/z$ 100 to 1,800, ion spray voltage of 5,500 V (ESI+) and −4500 V (ESI−), and a declustering potential of 80 V (ESI+) and −80 V (ESI−). An information-dependent acquisition (IDA) method was used to identify lipids, with the following conditions for MS analysis: collision energy of ±10, acquisition time of 250 ms. For MS/MS analysis, the conditions were the following: collision energy of ±45, collision energy spread of 25, ion release delay of 30, ion release width of 14, and acquisition time of 40 ms. The IDA switching criteria were set as follows: for ions greater than $m/z$ 300, which exceed 200 cps, exclude former target for 2 s, exclude isotopes within 1.5 Da, maximum candidate ions 20.

Before data analysis, raw MS data files were converted with the Reifycs Abf converter (v1.1) to the ABF file format. MS-DIAL (v4.20), with the FiehnO (VS68) database, was used to align the data and identify the different lipids (47–49). Further processing of the data was done with R version 4.0.2 (50).

The relative abundance of a specific lipid class versus the total relative abundance was used to roughly compare the ratio of each lipid class. The lipids were sorted into saturated and unsaturated lipids classes. Also, the lipids were sorted based on headgroups (DG, TG, PE, PI), and the ratio of each class was calculated.

**Lipopeptide preparation and treatment.** Peptides K$_4$ and E$_4$ were synthesized on a CEM Liberty Blue microwave-assisted peptide synthesizer using Fmoc chemistry. Piperidine at 20% in DMF was used as the deprotection agent. During coupling, N,N′-diisopropylcarbodiimide (DIC) was applied as the activator and Oxyma as the base. All peptides were synthesized on a Tentagel S RAM resin (0.22 mmol/g). The resin was allowed to swell for at least 15 min before synthesis started. For the coupling, 5 equivalents of amino acids (2.5 mL in DMF), DIC (1 mL in DMF), and Oxyma (0.5 mL in DMF) were added to the resin in the reaction vessel and were heated to 90°C for 4 min to facilitate the reaction. For deprotection, 20% piperidine (4 mL in DMF) was used and heated to 90°C for 1 min. Between deprotection and peptide coupling, the resin was washed three times using DMF. After peptide synthesis, a PEG$_4$ linker and cholesterol were coupled manually to the peptide on-resin. Each peptide (0.1 mmol) was reacted with 0.2 mM N$_3$-PEG$_4$-COOH by adding 0.4 mM O-(1H-6-chlorobenzotriazole-1-yl)-1,1,3,3-tetramethyluronium hexafluorophosphate (HCTU) and 0.6 mM N,N′-diisopropylethylamine (DIPEA) in 3 mL DMF. The reaction was performed at room temperature for 5 h. After thorough washing, 3 mL of 0.5 mM trimethylphosphine in a 1,4-dioxane–H$_2$O (6:1) mixture was added to the resin to reduce the azide group to an amine (overnight reaction). After reduction, the peptide was reacted with cholesteryl hemisuccinate (0.3 mmol) in DMF by adding 0.4 mM HCTU and 0.6 mmol DIPEA. The reaction was performed at room temperature for 3 h. Lipopeptides were cleaved from the resin using 3 mL of a trifluoroacetic acid (TFA)-triisopropylsilane (97.5:2.5) mixture and shaking for 50 min. After cleavage, the crude lipopeptides were precipitated by pouring into 45 mL of −20°C diethyl ether–n-hexane (1:1) and isolated by centrifugation. The pellet of the lipopeptides was redissolved by adding 20 mL H$_2$O containing 10% acetonitrile and freeze-dried to yield a white powder. Lipopeptides were purified with reversed-phase HPLC on a Shimazu system with two LC-8A pumps and an SPD-20A UV-Vis detector, equipped with a Vydac C$_4$ column (22-mm diameter, 250-mm length, 10-$\mu$m particle size). CPK$_4$ was purified using a linear gradient of 20% to 65% acetonitrile in water (with 0.1% TFA) with a 12-mL/min flow rate over 36 min. CPE$_4$ was purified using a linear gradient of 20% to 75% acetonitrile in water (with 0.1% TFA) with a 12-mL/min flow rate over 36 min. After high-performance LC (HPLC) purification, all peptides were lyophilized and yielded white powders.

For the fluo-K$_4$ and fluo-E$_4$ synthesis, two additional glycine residues were coupled to the N terminus of the peptides on resin, before the dye was manually coupled by adding 3 mL DMF containing 0.2 mmol 5(6)-carboxyfluorescein, 0.4 mmol HCTU, and 0.6 mmol DIPEA. The reaction was left at room temperature overnight. The fluo-K$_4$ and fluo-E$_4$ were cleaved from the resin using 3 mL of a TFA-triisopropylsilane-H$_2$O (97.5:2.5) mixture and shaking for 1.5 h. After cleavage, the crude lipopeptides were precipitated by pouring into 45 mL of −20°C diethyl ether and isolated by centrifugation. The pellet of the lipopeptides was redissolved by adding 20 mL H$_2$O containing 10% acetonitrile and freeze-dried to yield a white powder. Fluo-K$_4$ and fluo-E$_4$ were purified using the same HPLC described above equipped with a Kinetix Evo C$_{18}$ column (21.2-mm diameter, 150-mm length, 5-$\mu$m particle size). For the fluo-K4, a linear gradient of 20% to 45% acetonitrile in water (with 0.1% TFA) with a 12-mL/min flow rate over 28 min was used. For fluo-E4, a linear gradient of 20% to 55% was used. After HPLC purification, all peptides were lyophilized and yielded orange powders. The purity of all peptides was determined by LC-MS (see Table S2 in the supplemental material). The structure of all peptides used in this study can be found in Fig. S5. Treatment of cultures with different peptides was done by adding externally to cells suspended in P-buffer and incubating for 30 min at 30°C at 100 rpm. Excess peptide was washed by centrifugation.

**L-form membrane labeling.** Wild-type L-forms ($3 \times 10^8$) were suspended in 1 mL of P-buffer. A 10-$\mu$L aliquot of CPK$_4$ or CPE$_4$ (10 mM in DMSO) was added to the L-form suspension, to a final concentration of 100 $\mu$M. After 30 min of incubation at 30°C with shaking at 100 rpm, the L-forms were washed two times by centrifugation using P-buffer. The L-forms were then suspended in 900 $\mu$L P-buffer, and 100 $\mu$L of fluo-K$_4$ or fluo-E$_4$ (200 $\mu$M in P-buffer) was added to a final concentration of 20 $\mu$M. After 5 min of incubation, the L-forms were washed three times using P-buffer to get rid of the free fluorescent lipopeptides. For control experiments, fluo-K$_4$ or fluo-E$_4$ was added to non-lipopeptide-modified L-forms and incubated for 5 min. L-form imaging was performed on a Leica SP8 confocal microscope. Excitation was at 488 nm, and emission was determined at 500 to 550 nm.

**Peptide-induced L-form fusion.** Strains AG and HR were grown individually from culture stocks in 20 mL LPB containing the relevant antibiotic. Grown cultures were then centrifuged to remove spent medium containing antibiotic(s) and washed with P-buffer twice. The pellet was finally resuspended in 2 to 3 mL of P-buffer containing DNase I (1 mg/mL), and the density was adjusted to an $OD_{600}$ of 0.6. Peptides were added at required concentrations to 1-mL cultures of individual strains AG and HR. Cultures were then incubated for 30 min at 30°C with shaking at 100 rpm. Excess and unbound peptide was removed via centrifugation, and the pellet was resuspended in 1 mL P buffer containing DNase I. Both strains were then mixed in equal volumes (200 $\mu$L) in a fresh microcentrifuge tube and mixed gently, followed by incubation at room temperature for 10 min. Depending on the treatment, cultures were centrifuged followed by treatment with PEG1000 or simply centrifuged. The pellet was resuspended in 100 $\mu$L of P-buffer with DNase I, and serial dilutions were subsequently plated on LPMA with both antibiotics. Controls were also plated on the same medium as 100-$\mu$L monocultures of each strain to test for cross-resistance and as 100 $\mu$L of a 1:1 mix of each strain without fusion. All plates were incubated for 3 days, after which CFU were calculated to determine the fusion efficiency. Efficiency was quantified as CFU per milliliter on double antibiotic selection media normalized to the CFU per milliliter of monocultures grown on single antibiotic selection media.

**Statistical analysis and graphs.** Statistical analysis of all data sets was done in R version 3.6.1 (50) using built-in packages. The specific tests performed are mentioned in the results and Figure legends. All graphs were produced using the package ggplot2 (51).

## SUPPLEMENTAL MATERIAL

Supplemental material is available online only.

**SUPPLEMENTAL FILE 1**, AVI file, 9.4 MB.
**SUPPLEMENTAL FILE 2**, AVI file, 0.3 MB.
**SUPPLEMENTAL FILE 3**, AVI file, 3.2 MB.
**SUPPLEMENTAL FILE 4**, PDF file, 2.2 MB.

## ACKNOWLEDGMENTS

We thank members of the Claessen lab and Kros lab for fruitful discussions and suggestions. We thank Jaco Lugthart (Kros lab) for his contributions in the early phase of the project. We also thank 2 anonymous reviewers for critical comments that improved the manuscript. S.S. acknowledges the NWA startimpulse (Origins Centre) for funding.

S.S., D.C., and A.K. designed the project. S.S. performed all experiments. S.S. and M.S. performed peptide fusion experiments. M.-J.S. prepared all lipopeptides and did microscopy for lipopeptide docking experiments. B.C. prepared the cell wall-deficient line of *K. viridifaciens* used in the study. R.J.E.D. and M.G. performed the membrane lipid analysis. S.S., D.R., D.C., and A.K. acquired funding. S.S. wrote the first draft, followed by revisions from all authors. All authors approved the final manuscript.

We have no competing interests.

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
