## [Reviewer comments · Microbiology Spectrum]

Microbiology Spectrum

Generating heterokaryotic cells via targeted cell-cell fusion in wall-deficient bacteria

Shraddha Shitut, Meng-Jie Shen, Bart Claushuis, Rico Derks, Martin Giera, Daniel Rozen, Dennis Claessen, and A Kros

Corresponding Author(s): Shraddha Shitut, Leiden University

Review Timeline:

Submission Date:

May 15, 2022

Accepted:

June 22, 2022

Editor: Ilana Kolodkin-Gal

Reviewer(s): The reviewers have opted to remain anonymous.

Transaction Report:

DOI: <https://doi.org/10.1128/spectrum.01693-22>

We thank the reviewers for their positive remarks and constructive suggestions. We have addressed all raised concerns in a revised version of the manuscript and included additional supporting data. Furthermore, we have changed parts of the manuscript and changed the focus according to suggestions of reviewer #1. Below is a detailed response to each comment/concern.

Reviewer#1:

1. The authors should provide a better justification for why they consider their experimental procedure - besides being an interesting proof of principle that L forms can relatively easily be fused - to be relevant for understanding protocell fusion. I believe that the work should have been conducted under prebiotically more plausible conditions to justify this hypothesis. In fact, the experimental approach used in this study, although biophysically interesting, seems to be prebiotically irrelevant. We have no reason to believe that PEG, or PEG at the concentrations used in this study, or peptide docking strands existed on primitive Earth. Moreover, the authors achieved fusion by centrifuging cells at 1000 -6000 xg, which is another unlikely scenario in natural environments.

The reviewer raises a valid point regarding our experiments and conditions of "primitive Earth". We think that the similarities between L-forms and vesicles or protocells is interesting and allows to test certain hypothesis, which so far have received little attention. However, it is difficult to recapitulate the unique environment during early life. Therefore, we have modified the introduction and toned down this comparison (Lines 40, Introduction part).

2. From a biophysical point of view, fusion efficiency and viability of fused cells has not been convincingly demonstrated, as I do not find the data in the figures clearly supporting the claims. In my opinion, more reliable quantitative data on cell viability and fusion efficiency could be obtained from flow cytometry rather than by fluorescence microscopy of these small organisms being close to the optical resolution limit. In particular, the viability of the fused cells is not evident from Figure 3. Most cells in the image did not show either growth (increase in size) or morphology change over 16h. Two cells specifically pointed in the image, appear to be undergoing lysis rather than reproduction. It would be helpful if authors provide better images of cells undergoing reproduction.

The two concerns pointed out here are – the method of quantifying fusion efficiency and visualization of cell viability after fusion.

For quantifying fusion efficiency, we have used the measure of colony forming units (CFU/ml) on media with both antibiotics. In the new supplementary figure 1 we show that only fused cells can grow on media with both antibiotics justifying the use of this measure. The suggestion by the reviewer of flow cytometry for quantifying the number of fused cells is possible but comes with potential hurdles. The first is that from our experiments as well as previous work on protoplast fusion we see the need of a solid stable surface for the process of fusion itself (1, 2). Peptides and PEG play the role of bringing cells in proximity (3, 4) followed by the merging of membranes which potentially happens on the media with both antibiotics. If a culture consisting of the two strain types is treated with PEG or peptides and then run through a flow cytometer, we don't see the double labeled cells since the window of fusion has not yet arrived. Additionally, we risk the possibility of clumps or aggregates of cells (formed due to the fusogens) giving a false positive signal. Given these shortcomings we decided to stick

with the current fusion efficiency measurement since each colony represents a fusion event. We also explain this in the text (line 143-153).

The second concern regarding cell viability after fusion has been rectified through two changes. First in the revised Figure 2, we have included images of individual fused cells grown in liquid. These data convincingly show cells that express both eGFP and mCherry, indicating the presence of both chromosomes within the same cell. We also renewed Figure 3 with better timelapse images and the supplementary movie showing growth of fused cells. Please note that all our images/movies are consistent with earlier studies demonstrating L-form proliferation (Supplementary movie 1) (5, 6). Lastly, supplementary figure 2A shows growth of fused cells quantified by optical density measurements and show the similarity between densities of fused cultures with parental strains.

3. By working with a single species of bacteria, authors did not achieve the original goal of understanding the genome diversification due to cell fusion. The results presented here at best describes mixing of cytoplasmic contents.

We agree with the reviewer that the cells fused here have diversity only in one region of the chromosome where the markers have been introduced. The reason for using only one species for this study was to understand the process of fusion, the ease of genetic tractability and to standardize methods. Despite using one species, we did observe and quantify the modified antimicrobial activity in fusants compared to the parental strains (supplementary figure 4). These results give us the necessary base to test and hypothesize the result of inter-species fusion, which is the scope of a subsequent paper. We discussed the future possibilities on this study, i.e. interspecies fusion, in the discussion section (line 411-416).

4. In my opinion a crucial control experiment is missing. Presence of both GFP and mCherry within a cell was interpreted to be a result of cell fusion. Previous studies showed that growth and reproduction in L-forms is very leaky with loss of cytoplasmic constituents and L-forms are also known to take up DNA from the surroundings. Keeping this in mind, there appears to be another possibility for the presence of both GFP and mCherry in a cell, i.e., L-forms underwent lysis and released DNA into the surroundings. Some cells should have acquired plasmids for the surrounding media rather than by cell fusion. One can disregard this possibility by adding DNase into the growth media. Such a control experiment is missing.

We agree with the reviewer that this was a plausible outcome. Therefore, all fusion experiments were performed in the presence of DNase I (1 mg/ml), which has shown to be sufficient to degrade any free DNA (7). Furthermore, in a control experiment two cell types were mixed without fusion (i.e. no centrifugation, PEG or peptides were used/added) and plated on double antibiotic media. As expected, no colonies were observed (Supplementary Figure 1), confirming fusion is required for colony growth on double antibiotic selection media. In another control experiment we confirmed that colonies appearing on double antibiotic selection media are not a result of DNA uptake followed by recombination. For this, fusant colonies were replated on media without any antibiotics and imaged after 3 days of growth (supplementary figure 3). Here the segregation of the two chromosomes within cells was observed as regions of only green and red fluorescence. Pixel colocalization in these images resulted in a negative trend compared to the biomass growing on media with both antibiotics. We also explain this in the text (line 192-199).

5. A consistent story line is missing. Throughout the manuscript authors perplexed between different topics. In the abstract and introduction, authors explained why L-forms are a better model system for studying protocells and proposed the objective of understanding their genome evolution via cell fusion. The procedural approach is at odds with this narrative (point 1). Protocells or early life are barely mentioned in the discussion, which is mostly leaned towards applied aspects of biology, like targeted delivery of compounds into L-forms and treating urinary tract and mycobacterial infections.

I believe the work has some innovative aspects like using L-forms as proxy protocells. But considering the experimental approach and implications discussed by the authors, it would be better to largely rewrite the manuscript from the viewpoint of applied biology and highlight potential applications in biotechnology and infection biology.

We have changed the focus of the manuscript accordingly.

Minor comment: The authors forgot to describe the protocol of lipid analysis in the methods section. This should be fixed.

The lipid analysis protocol was added to the supplementary methods section.

Reviewer#2:

Minor comments and suggestions:

1) Please revise the manuscript (especially the abstract) to emphasise the fusion of L-forms was induced (by PEG or peptides). Otherwise, the readers might have an impression the L-form fusion was spontaneous.

We have now clarified this statement (see lines 45, 95).

2) Page 6, top paragraph (Fig 2A): How do we know the lysis actually occurred at speeds over 6000xg? This most likely is the case, but the experiments don't explicitly show that. Soften the language.

The statement has been modified (line 137).

3) In the same paragraph, to claim membrane composition of L-forms and protoplasts is different, the two have to be compared side by side. Soften the language to emphasise this is just a suggestion or carry out the experiments to compare membrane composition of L-forms and protoplasts.

We included this statement based on previous literature where protoplasts and L-forms of other *Streptomyces* species have been analyzed. However, we agree that this specific analysis has not been provided by us in the paper and so have revised those statements (line 141).

4) Genotype the fusants to further confirm on a genetic level the fusion has occurred. We have included results from a PCR analysis of genomic DNA purified from the parental strains (AG, HR) and two fusants (F1, F2) in Supplementary Figure 2B, which unambiguously show the presence of both markers. We also mention this in the text (line 189).

5) Page 8, paragraph 2: What do you mean by "casually confirm"? Either you confirm or you don't

The phrase we use is “causally confirm” by which we mean to identify the causative factor. This statement has been clarified (line 241).

6) Page 10, end of the paragraph 2: "... and the to the presence of antibiotics"? Typo? Statement corrected (line 305).

7) Page 12, beginning of paragraph 3: "...which are be sufficiently fluid"? Typo? Statement corrected. (line 365).

References

1. Peberdy JF. 1980. Protoplast fusion — a tool for genetic manipulation and breeding in industrial microorganisms. *Enzyme and Microbial Technology* 2:23–29.
2. Schaeffer P, Cami B, Hotchkiss R D. 1976. Fusion of bacterial protoplasts. *Proceedings of the National Academy of Sciences* 73:2151–2155.
3. MacDonald RI. 1985. Membrane fusion due to dehydration by polyethylene glycol, dextran, or sucrose. *Biochemistry* 24:4058–4066.
4. Wojcieszyn JW, Schlegel RA, Lumley-Sapanski K, Jacobson KA. 1983. Studies on the mechanism of polyethylene glycol-mediated cell fusion using fluorescent membrane and cytoplasmic probes. *Journal of Cell Biology* 96:151–159.
5. Mercier R, Kawai Y, Errington J. 2014. General principles for the formation and proliferation of a wall-free (L-form) state in bacteria. *eLife* 3:e04629.
6. Studer P, Staubli T, Wieser N, Wolf P, Schuppler M, Loessner MJ. 2016. Proliferation of *Listeria monocytogenes* L-form cells by formation of internal and external vesicles. *Nat Commun* 7:13631–13631.
7. Kapteijn R, Shitut S, Aschmann D, Zhang L, de Beer M, Daviran D, Rovers R, Akiva A, van Wezel GP, Kros A, Claessen D. 2022. DNA uptake by cell wall-deficient bacteria reveals a putative ancient macromolecule uptake mechanism. *bioRxiv* 2022.01.27.478057.

June 22, 2022

Dr. Shraddha Shitut
Leiden University
Leiden 2333 BE
Netherlands

Re: Spectrum01693-22 (Generating heterokaryotic cells via targeted cell-cell fusion in wall-deficient bacteria)

Dear Dr. Shraddha Shitut:

Your manuscript has been accepted, and I am forwarding it to the ASM Journals Department for publication. You will be notified when your proofs are ready to be viewed.

Sincerely,

Ilana Kolodkin-Gal
Editor, Microbiology Spectrum

Journals Department
Supplemental Movie S3: Accept
Supplemental Material: Accept
Supplemental Movie S2: Accept
Supplemental Movie S1: Accept